# A ubiquitous GC content signature underlies multimodal mRNA regulation by DDX3X

Ziad Jowhar [1,2,3,11], Albert Xu [1,2,3,11], Srivats Venkataramanan [1], Francesco Dossena [4], Mariah L Hoye [5], Debra L Silver [5,6,7,8,9], Stephen N Floor [1,10 ✉] & Lorenzo Calviello [4 ✉]

## Abstract

**The road from transcription to protein synthesis is paved with many obstacles, allowing for several modes of post-transcriptional regulation of gene expression. A fundamental player in mRNA biology is DDX3X, an RNA binding protein that canonically regulates mRNA translation. By monitoring dynamics of mRNA abundance and translation following DDX3X depletion, we observe stabilization of translationally suppressed mRNAs. We use interpretable statistical learning models to uncover GC content in the coding sequence as the major feature underlying RNA stabilization. This result corroborates GC content-related mRNA regulation detectable in other studies, including hundreds of ENCODE datasets and recent work focusing on mRNA dynamics in the cell cycle. We provide further evidence for mRNA stabilization by detailed analysis of RNA-seq profiles in hundreds of samples, including a *Ddx3x* conditional knockout mouse model exhibiting cell cycle and neurogenesis defects. Our study identifies a ubiquitous feature underlying mRNA regulation and highlights the importance of quantifying multiple steps of the gene expression cascade, where RNA abundance and protein production are often uncoupled.**

**Keywords** Transcriptomics; DDX3X; mRNA Biology; Translation; Computational Biology
**Subject Categories** Chromatin, Transcription & Genomics; RNA Biology

## Introduction

The cytoplasmic fate of RNA molecules is affected by their subcellular localization, RNA binding partners, and engagement with the ribosomal machinery. These aspects are strongly interconnected (Shoemaker and Green, 2012), which poses a great challenge, as it increases the number of variables and experimental approaches needed to answer many questions in mRNA biology.

To this end, many protocols couple biochemical isolation, or metabolic labeling, of RNA with high throughput sequencing technologies, thus providing a snapshot of the transcriptome at specific stages of the mRNA life cycle, with high throughput and sensitivity. For example, high-throughput sequencing protocols, when coupled to ribosome isolation, such as in Ribo-seq (Ingolia et al, 2012), metabolic labeling strategies in SLAM-seq (Herzog et al, 2017), immunoprecipitation of RNA binding proteins (RBP) as in CLIP-seq (Hafner et al, 2021) and many others, have shed light on many regulatory mechanisms pertaining to different aspects of RNA biology.

DDX3X is a multifunctional RNA helicase that is highly expressed in many tissues and able to unwind structured RNA to influence cytoplasmic post-transcriptional gene regulation (Sharma and Jankowsky, 2014). Together with its ability to bind initiating ribosomes, DDX3X has been often described as a translation regulator, specifically promoting translation of RNA with structured 5'UTRs (Oh et al, 2016; Calviello et al, 2021). However, as mentioned above, cytoplasmic processes like translation or mRNA decay are intertwined, and connection between the two processes encompass different molecular mechanisms, such as mRNA surveillance mechanisms like nonsense-mediated decay (NMD) (Chang et al, 2007), ribosome-collision dependent mRNA cleavage (D'Orazio et al, 2019), and others. To understand when and how such processes are coupled, it is important to study the dynamics of such mechanisms. For instance, it has been proposed that miRNA can first trigger translation suppression and then mRNA deadenylation and decapping leading to RNA degradation (Bazzini et al, 2012).

Mutations in *DDX3X* are associated with a variety of human diseases including cancers and developmental delay (Gadek et al, 2023). Variant types are disease selective in *DDX3X*, with cancers ranging from primarily loss-of-function alleles in NK-TCL and other blood cancers to nearly exclusively missense variants in medulloblastoma (Lennox et al, 2020). In *DDX3X* syndrome, missense variants are phenotypically more severe than loss-of-function. Previously, we used functional genomics approaches to identify mechanistic differences between the depletion of DDX3X and the expression of missense variants (Calviello et al, 2021). We

---

[1]Department of Cell and Tissue Biology, UCSF, San Francisco, USA. [2]Medical Scientist Training Program, University of California, San Francisco, San Francisco, CA 94158, USA. [3]Biomedical Sciences Graduate Program, University of California, San Francisco, CA 94158, USA. [4]Human Technopole, Milan, Italy. [5]Department of Molecular Genetics and Microbiology, Duke University Medical Center, Durham, USA. [6]Department of Cell Biology, Duke University Medical Center, Durham, USA. [7]Duke Regeneration Center, Duke University Medical Center, Durham, USA. [8]Department of Neurobiology, Duke University Medical Center, Durham, USA. [9]Duke Institute for Brain Sciences, Duke University Medical Center, Durham, USA. [10]Helen Diller Family Comprehensive Cancer Center, San Francisco, USA. [11]These authors contributed equally: Ziad Jowhar, Albert Xu. ✉E-mail: Stephen.Floor@ucsf.edu; Lorenzo.Calviello@fht.org

found that *DDX3X* missense variants predominantly affect ribosome occupancy while DDX3X depletion impacts both ribosome occupancy and RNA levels. However, it is unclear whether the changes in RNA levels constituted a cellular response to translation suppression, often described as "buffering" or "offsetting" (Ingolia, 2016; Lorent et al, 2019).

mRNA regulation has been linked to neurogenesis during development, where multiple RNA binding factors, including DDX3X, ensure correct protein synthesis as cells transition between different fates and states (Hoye et al, 2022). To that end, it is important to think about the dynamics of gene expression, as complex dynamics of cell proliferation and differentiation ensure correct developmental patterning.

In order to access such complex interplays of a multitude of factors that shape gene expression, large-scale consortia have provided a great resource for investigations into gene regulation. While historically devoted to promoting investigation into transcriptional regulation, recent efforts started to provide precious information into post-transcriptional mechanisms, with hundreds of RBPs profiled in terms of both binding and function, by means of CLIP-seq, and knockdown followed by RNA-seq (Van Nostrand et al, 2020a). As in biology many molecular processes are interconnected, large-scale datasets and data amenable to re-analysis are at the very heart of many research efforts (Hon and Carninci, 2020).

Here, we identify how inactivation of DDX3X evolves over time to lead to acute and long-term changes to post-transcriptional gene regulation. We here employ different analytical approaches applied to newly generated experimental data and many previously published studies related to mRNA regulation, to show that GC content is associated with mRNA stability changes following DDX3X depletion. Our analyses indicate that this effect is widespread and is associated with cell cycle changes in mRNA regulation, including RNA stability. This further reinforces roles for DDX3X in RNA stability in addition to translation. Together, our work represents a significant advancement in the understanding of a fundamental regulator, which sits at the very heart of the gene expression cascade.

# Results

## Time-resolved gene expression regulation by DDX3X

To characterize the dynamics of DDX3X-dependent changes in the gene expression cascade, we employed a previously validated auxin-degron system to efficiently deplete DDX3X protein in the HCT116 colorectal cancer cell line (Venkataramanan et al, 2021), where we found near-complete rescue of gene expression changes by DDX3X expression, thus being able to use this tool to monitor DDX3X-dependent changes. We profiled RNA levels and translation using RNA-seq and Ribo-seq along a time-course of DDX3X depletion, at 4, 8, 16, 24 and 48 h after auxin or DMSO control treatment. (Fig. 1A). Efficiency of DDX3X depletion, together with quality control and general statistics of the generated libraries, can be found in Appendix Fig. S1 and Dataset EV1. As expected, the number of differentially expressed genes increased along the time-course, with most changes supporting the role of DDX3X as a positive regulator of translation (Fig. 1B). Changes in translation

were negatively correlated with changes in mRNA levels, which together contributed to many changes in Translation Efficiency (TE), calculated using Ribo-seq changes given RNA-seq changes (Methods). At a closer look, we observed how "TE_down" mRNAs undergo translation suppression in the early time point after DDX3X depletion, with their mRNA levels increasing in the later time points (Fig. 1C). The opposite behavior is observed for "TE_up" mRNAs, exhibiting higher ribosome occupancy first, and lower mRNA levels later. Such behavior was more evident when showing time-point specific changes and binning mRNAs in a 2D grid on the Ribo-seq/RNA-seq coordinate plane (Fig. 1D, Methods), which highlighted a common regulatory mode, with early translation regulation followed by changes in mRNA levels.

This analysis shows the time-resolved dynamics of mRNA regulation by DDX3X, with hundreds of mRNAs changing in their steady-state levels albeit showing the opposite directionality in translation rates.

## Translation suppression by DDX3X is coupled with mRNA stabilization

Changes in transcript levels can result from changes in transcription rates or post-transcriptional regulation. To identify the relative contribution of different processes to RNA levels, we used our time-course dataset to calculate changes in transcription, processing and stability using *INSPEcT* (De Pretis et al, 2015). *INSPEcT* uses the proportion of intronic versus exonic reads to identify nascent vs. mature transcripts, and it solves a system of ordinary differential equations (ODEs) to infer rates of RNA synthesis, processing and decay. Compared to non-regulated mRNAs, regulated mRNAs showed modest changes in transcription rates, suggesting transcription changes are not the major contributor to RNA level changes following DDX3 depletion. In contrast, we found more pronounced changes in mRNA stability as evidenced by TE down transcripts (Fig. 2A). As our initial RNA-seq protocol was not designed to capture pre-mRNA molecules, we validated our estimated mRNA stability changes by employing the 4sU metabolic labeling SLAM-seq protocol (Herzog et al, 2017) in our degron system after 8 h of DDX3X depletion, in a way to detect changes in mRNA stability at early time points. Briefly, cells were incubated with 4sU to comprehensively label transcribed RNAs, and their abundance was followed after 8 h of DDX3X degron activation, using DMSO as control. 4sU treatment induces T > C conversions in the sequenced cDNA molecules, which can be used to monitor mRNA stability changes after a uridine chase, as shown in Fig. 2B. As expected, we observed a drastic drop in T > C harboring reads after the chase, which reflects mRNA decay rates (Appendix Fig. S2). As shown in Fig. 2B, after a labeling time of 24 h, the percentage of reads harboring T > C mutations was different for the regulated categories (Methods) after only 8 h of degron induction, confirming the stabilization of translationally suppressed mRNAs upon DDX3X depletion. While the modest depth and resolution of our SLAM-seq dataset (Appendix Fig. S2) couldn't allow for more detailed insights on mRNA changes, it represented an important validation of mRNA stability regulation by DDX3X. In addition, we profiled RNA abundance via qPCR, combining our DDX3X degron system with Actinomycin D (ActD) treatment, to measure RNA stability changes. We selected few target mRNAs: *JUND* was identified in our data as a stabilized RNA, while *EIF2A* as degraded.

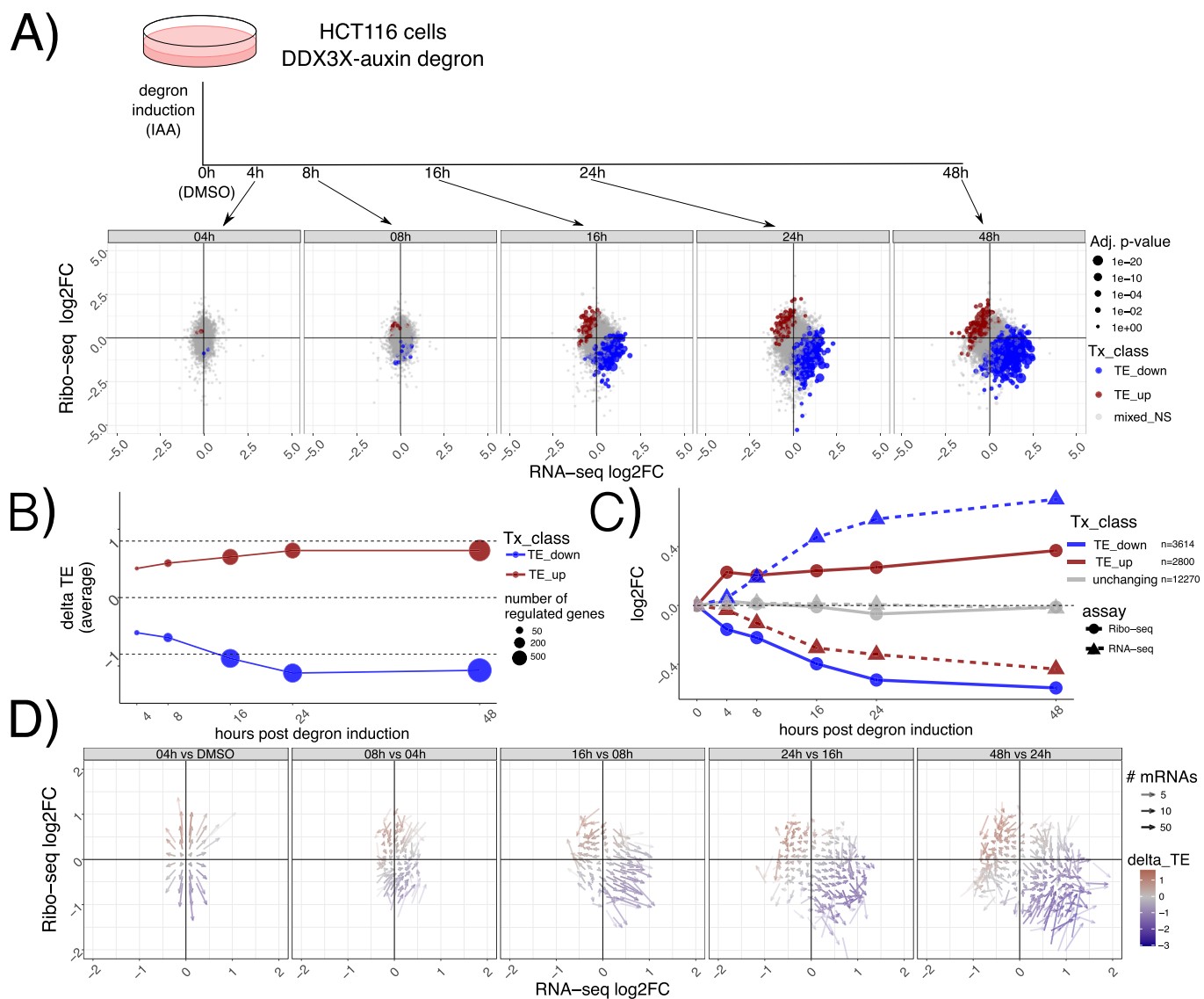

**Figure 1. Dynamics of mRNA regulation by DDX3X.**

In (A) a description of the experimental design. Below Ribo-seq and RNA-seq fold changes at different time points. Different regulated classes are shown in different colors. The size of the dots indicates the adjusted *P*-values from a differential translation efficiency test using *DESeq2* (Methods). NS: not significant. In (B) average delta TE values (differences in TE values) for each class along the time course. The size of the dots indicates the number of significantly changing mRNAs. (C) progression along the time course for mRNA regulated 48 h post degron induction. RNA-seq and Ribo-seq fold changes are shown at each time point. (D) Differences in Ribo-seq or RNA-seq fold changes between each time point and the previous one, shown as a vector plot. Magnitude of changes shown as a color gradient, while transparency of the vectors indicates the number of mRNAs in each coordinate bin (Methods).

*RACK1*, *LGALS1*, and *PFN1* were used as controls to normalize with via RT-PCR with Taq-man probes. *JUND* RNA was stabilized after 24 h with knockdown of DDX3 and ActD treatment (Appendix Fig. S3A); *EIF2A* RNA was more degraded after 24 h with knockdown of DDX3 and ActD (Appendix Fig. S3B). These results show an overall good agreement between the qPCR and the sequencing-based assays, despite the difficulty arising from choosing control mRNAs and the modest fold changes observed in the sequencing data.

By profiling ribosome occupancy, steady state transcript levels, and mRNA decay, this analysis shows that DDX3X depletion triggers multiple modes of post-transcriptional regulation,

involving translation suppression and a subsequent wave of mRNA stabilization.

## GC-rich coding sequences underlie mRNA regulation by DDX3X

With hundreds of mRNAs post-transcriptionally regulated after DDX3X depletion, we aimed to identify specific features belonging to up- or downregulated targets. We therefore built regression models to quantitatively predict levels of TE changes (Methods, Dataset EV2). We used different biophysical properties of genes and mRNAs, (e.g. length and GC content) and several gene and

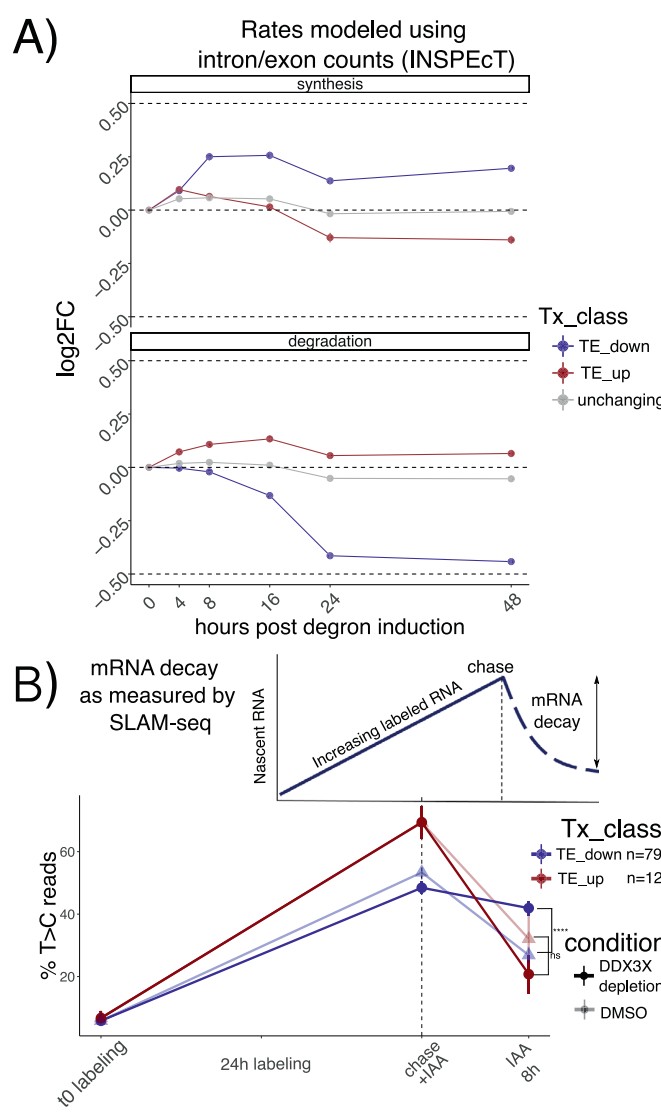

**Figure 2. Stabilization of untranslated mRNAs.**

(A) Synthesis and decay as inferred by *INSPEcT*: different regulated classes in different colors along the time course. Log2FC of estimated rates with respect to control are shown on the y axis. n of mRNAs: TE_down = 1809, TE_up = 1401; unchanging = 6136. Error bars represent the standard error of the mean. (B) Schematic of a SLAM-seq experiment (above). Real data shown at the bottom: percentage of T > C-containing reads on the *y* axis after labeling and chase. DDX3X degron (using DMSO as a control) was triggered together with the chase reaction to monitor differences in decay rates upon DDX3X depletion. Significance values from a one-sided Wilcoxon test, using mRNAs significantly changing in both translation (from RNA-seq and Ribo-seq) and stability (from SLAM-seq, Methods), showing the following symbols: "ns" = *p* > 0.1; "." = *p* <= 0.1; "*" = *p* <= 0.05; "***" = *p* <= 0.01; "****" = *p* <= 0.001; "*****" = *p* <= 0.0001. Error bars represent the standard error of the mean.

transcript features (e.g. introns, 3'UTR, etc., Methods) as features for a Random Forest regression model. Given the extensive literature on codon-mediated mRNA stability regulation, we added codon frequencies and previously validated codon optimality calculations (Medina-Muñoz et al, 2021). Also, we added measured GC-content at 1st, 2nd or 3rd codon position, as it was recently shown to potentially play a role in mRNA stability regulation

(Courel et al, 2019; Hia et al, 2019). In addition, to pinpoint features predictive of mRNA stability changes rather than translation changes exclusively, we divided transcripts according to their position in the Ribo-seq/RNA-seq coordinate system, to capture mRNAs where changes between assays agreed or not (Fig. 3A, Methods). Interestingly, the categories differed in their DDX3X binding pattern (Appendix Fig. S4): re-analysis of our previously published PAR-CLIP data showed how stabilized targes (x,-xy groups) have a lower T > C conversion signal in their 5'UTRs, and a higher signal in CDS peaks, with the opposite being true for targets on the Ribo-seq axis (y group). This analysis suggests that stabilized mRNAs might be regulated differently than "canonical" translationally suppressed targets.

As shown in Fig. 3B, the Random Forest model predicted TE changes with high precision, especially in cases where mRNA stability and translation were anti-correlated (-xy group). In addition, this model calculated the predictive power of each input feature (Fig. 3C, Methods), which highlighted GC content in the coding sequence (which we will refer to as *GCcds*) as the most important feature. Feature selection is a very important method to select predictive features, especially when facing high levels of multicollinearity (Appendix Fig. S5). To validate the results from the Random Forest regression, we used Lasso regression (Methods), another well-known method for feature selection. Results from the Lasso regression were similar, and also identified GC content in the coding sequence as the most relevant feature in predicting TE changes (Appendix Fig. S6). GC content in the CDS remained the top predictor when using additional features, such as GC content in different sections of the CDS, or amino acid frequencies (Appendix Fig. S7).

In the light of these results, we tested whether *GCcds* was associated with the DDX3X-dependent transcriptome dynamics reported above. As shown in Fig. 3D, mRNAs partitioned on the Ribo-seq/RNA-seq coordinate system based on their *GCcds* value. Moreover, stability values from both *INSPEcT* and SLAM-seq partitioned according to *GCcds* values (Fig. 3E,F). A similar, albeit weaker, separation was observed for predicted transcription and processing rates (Appendix Fig. S8).

By using multiple analytical approaches, we here show how *GCcds*, not just GC content in general, or in other sections of the transcriptome, is a predominant feature of stabilized, yet untranslated, mRNAs following DDX3X depletion.

## GC content in the coding sequence is a ubiquitous signal in mRNA regulation

Given the extensive connections between different aspects of mRNA regulation by thousands of regulators, we tested the breadth of the influence of features such as *GCcds* in other studies of RNA regulators. We re-analyzed >2000 RNA-seq samples (Methods) from the recent ENCODE RBPome (Van Nostrand et al, 2020a) study encompassing >200 RBP knockdowns, and performed differential analysis followed by predictive modeling using the same methods and features as described in the previous section, this time aiming at predicting changes in mRNA levels (Fig. 4A).

We first grouped datasets according to knockdown efficiency, which varied according to knockdown method and cell line (Appendix Fig. S9, Methods). We selected the sample with the highest knockdown efficiency for each RBP and called feature importance using our analytical pipeline. Predictive power of our

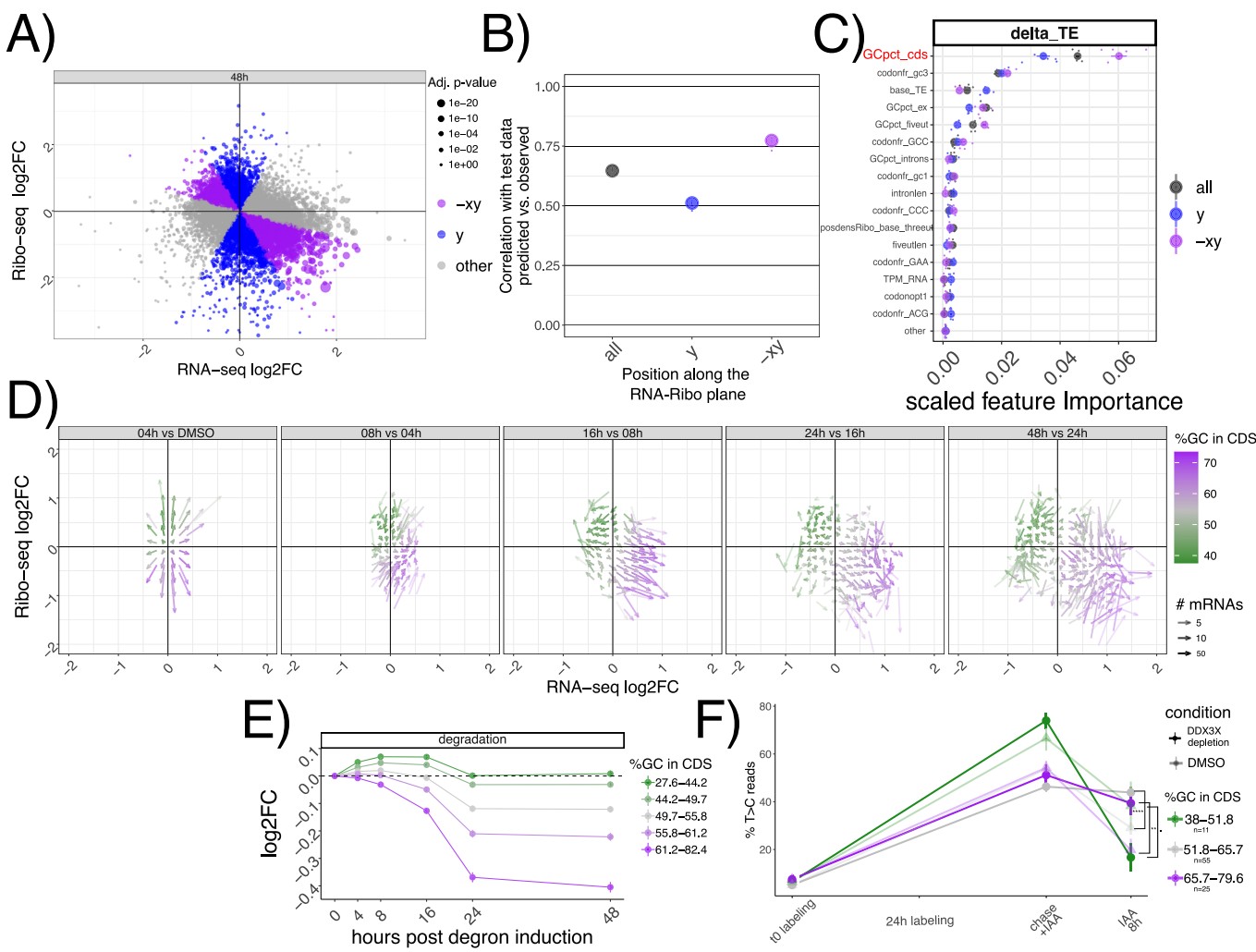

**Figure 3. GC content in the coding sequence predicts regulation by DDX3X.**

(A) Classification of different mRNAs according to their change in mRNA levels or translation. The size of the dots represents statistical significance from a differential translation efficiency test, as in Fig. 1A. In (B) model performance (correlation between predicted vs. real values) on unseen test data of the random forest regression model for transcript classes as defined in (A). Performance shown for each round of cross-validation, $n = 5$; error bars represent the standard error of the mean. (C) Predictive power of most informative features, with their importance values (Methods) plotted on the x axis. Features pertaining to GC content in different section of transcripts (*GCpct**), baseline translation levels (*base_TE*), codon frequencies (*codonfr**), positional read density (*posdens**), and length features (*intronlen*) are displayed. Performance shown for each round of cross-validation, $n = 5$. Error bars represent the standard error of the mean. (D) Vector plot as in Fig. 1D, highlighting *GCcds* values. Inferred degradation rate in (E), and SLAM-seq profiles in (F), for mRNAs partitioned by *GCcds* values. Number of mRNAs for panel (E): "27.6-44.2" = 1861; "44.2-49.7" = 1892; "49.7-55.8" = 1854; "55.8-61.2" = 1847; "61.2-82.4" = 1892. Error bars represent the standard error of the mean. Significance values in (F) from a one-sided Wilcoxon test, using the same mRNAs from Fig. 2B, showing the following symbols: "ns" = $p > 0.1$; "." = $p <= 0.1$; "*" = $p <= 0.05$; "**" = $p <= 0.01$; "***" = $p <= 0.001$; "****" = $p <= 0.0001$. Error bars represent the standard error of the mean.

Random Forest regression strategy varied across different datasets (Fig. 4B). Once again, the strongest predictor of mRNA changes was *GCcds*, whose predictive power dominated over other variables (Fig. 4C, Appendix Fig. S10). As expected, changes upon DDX3X knockdown in the ENCODE data also exhibited a clear dependency over *GCcds* (Fig. 4D), albeit to a lower degree compared to our degron dataset, likely due to differences in DDX3X depletion strategies and, importantly, to our translation profiling dataset, which allowed us to distinguish between specific classes (i.e. "TE_down") of regulated mRNAs (Discussion).

Given the widespread relevance of *GCcds* as a predictor of post-transcriptionally regulated targets, we reasoned that a major

cellular process might mediate the observed mRNA changes. We re-analyzed data from a recent study (Krenning et al, 2022) focused on mRNA clearance during cell cycle re-entry, where the authors used a FUCCI (fluorescent, ubiquitination-based cell-cycle indicators) cell system coupling RNA labeling, scRNA-seq and single-molecule imaging techniques to find extensive decay differences among different transcripts, potentially related to poly-A tail mediated regulation. Despite a lower throughput when compared to sequencing-based experiments, kinetic parameters estimated from their data (exemplified in the decay curve in Fig. 4E) showed significant differences when partitioned by *GCcds* values (Fig. 4E). mRNAs rich in *GCcds* showed lower half-life

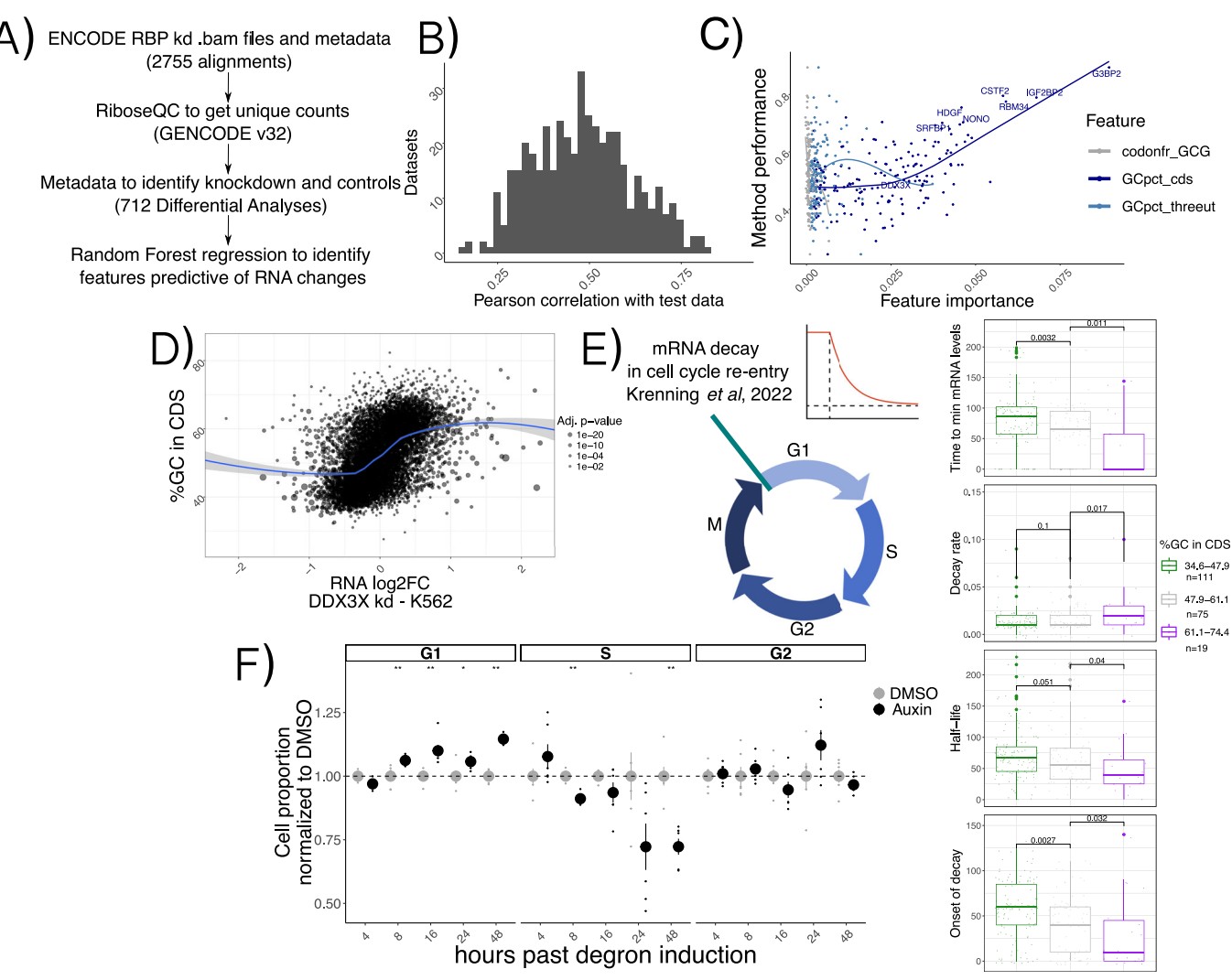

**Figure 4. A ubiquitous feature in mRNA regulation.**

(**A**) Schema describing the ENCODE analysis strategy. (**B**) Histogram representing overall model performance across datasets. (**C**) Model performance (spearman correlation between predicted and real values on unseen test data) on the y axis, with importance of 3 example features variables (indicating their predictive value) on the x axis. Top knockdown experiments, together with DDX3X, are show with labels. Data shown are from shRNA KD experiments in K562 cells. The linear relationship between GCcds importance and model performance indicates its relevance as the top predictor of RNA changes in dozens of datasets. (**D**) mRNA level changes against GCcds values in a DDX3X knockdown experiment in the ENCODE dataset; the size of the dots represents statistical significance of the differential expression test vs. control samples, as defined by DESeq2. (**E**) Schematics of the cell cycle data used. Values for different kinetic parameters were partitioned according to GCcds values of their mRNAs and tested for significant differences. For all box plots, in each box, the central black line indicates the median, and the upper and lower edges denote the 25th and 75th percentiles. Whiskers represent 1.5 times the interquartile range from the box hinge. The number of mRNAs analyzed is shown in the legend box. (**F**) Normalized cell proportion (obtained by dividing cell percentages between Auxin treatment and DMSO) in different stages of the cell cycle along the degron time course. An increase in G1 and decrease in S phase can be observed at later time points. Significance values calculated from a Wilcoxon two-sided test. Data from two experiments performed in triplicate, $n = 6$. Error bars represent the standard error of the mean.

values, and fast decay kinetics at cell cycle re-entry, with the opposite trend exhibited by mRNAs poor in GC content in their coding sequence. Motivated by this finding, we decided to investigate differences in cell cycle dynamics in our degron system, by using 5-ethynyl-2'-deoxyuridine (EdU) incorporation followed by FACS analysis (Methods, Appendix Fig. S11). As shown in Fig. 4F and Appendix Fig. S12, DDX3X depletion resulted in cells staying more in G1 and less in S phase when compared to controls, throughout the time course.

By re-analysis of thousands of RNA-seq samples, these results show the prevalence of *GCcds* in post-transcriptional regulation

and RBP functions, with a potential role for cell-cycle dependent mRNA dynamics in shaping such a regulatory phenomenon.

## A shift in 5'-3' RNA-coverage as a hallmark of mRNA stabilization

In addition to gene-level aggregate measures of abundance, we investigated whether changes in decay could be identified by taking advantage of the high resolution of RNA-seq experiments across gene bodies, which has previously been employed to inform about mRNA decay (Courel et al, 2019). We leveraged our time-resolved degron

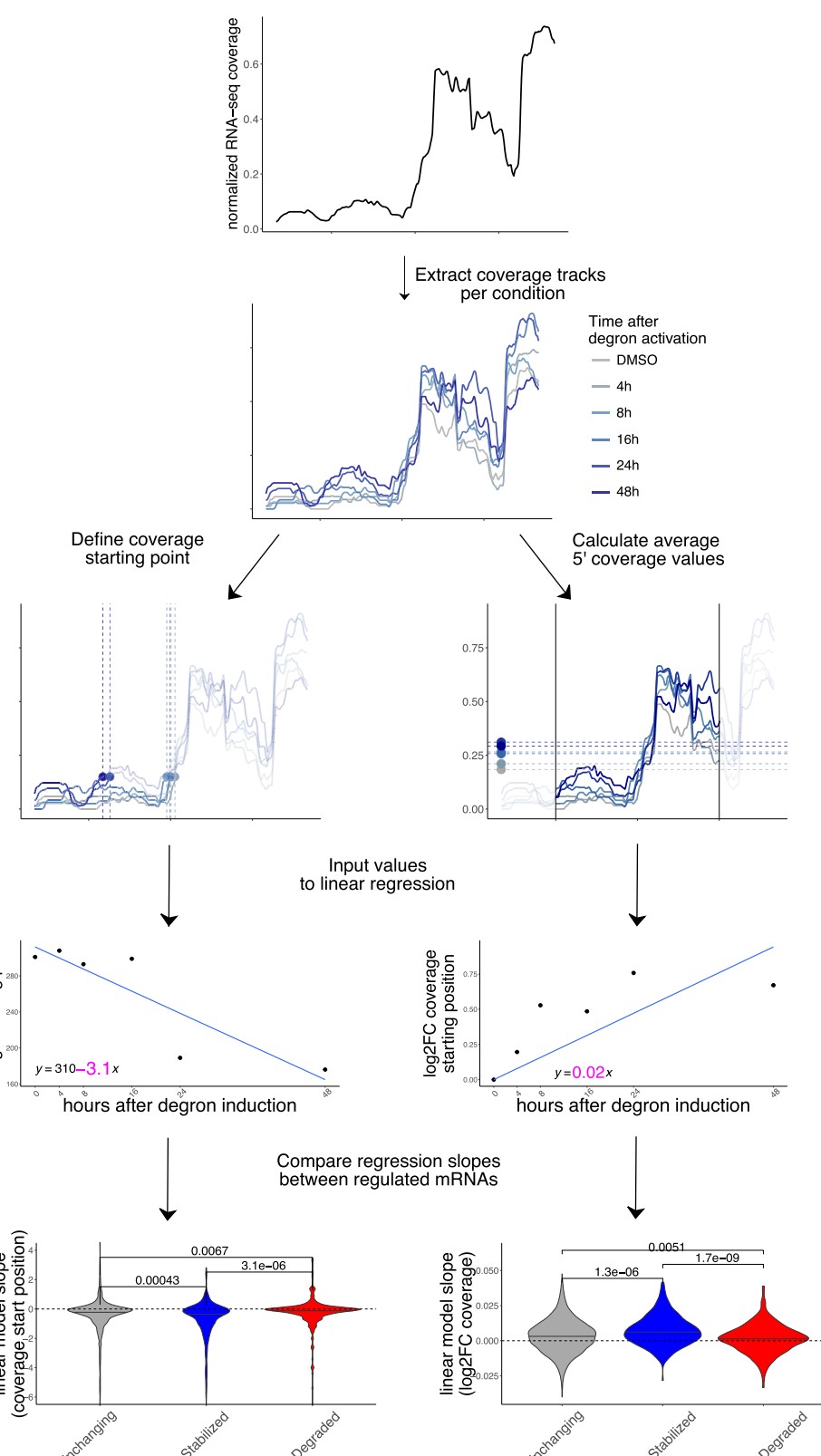

**Figure 5. Coverage analysis of regulated mRNAs reveals changes in 5'-3' decay.**

Coverage analysis strategy in the degron dataset using a practical example (CSRNP2 gene): coverage starting point is first identified using pooled data, then coverage tracks for each experiment are extracted. Coverage starting points (in transcript coordinates) and coverage values (log2FC to DMSO) are calculated for each time point and used as input to a linear model. The beta coefficient (shown in pink) for each model is then extracted for each mRNA and values are compared across different classes (stabilized vs unchanging vs degraded). More details are available in the Methods section. *P*-values from one-sided Wilcoxon test.

dataset to investigate changes in 5'-3' coverage, a known hallmark of RNA degradation often employed to verify overall integrity of cellular mRNAs or to estimate transcript-level decay. We calculated 2 different metrics, using the strategy illustrated in Fig. 5.

Initially, we pooled samples to identify the first position at 15% of the maximum coverage (Methods). We then calculated such position for each time point. We then used coverage starting points as input for linear regression. The regression coefficient was calculated for the top 250 stabilized and degraded mRNAs, alongside 1500 control transcripts. As shown in Fig. 5, coverage values on stabilized mRNAs started as an earlier position in the transcripts, with moderate albeit significant differences between categories, indicating a lower 5'-3' decay along the DDX3X degron time course. The opposite trend was observed for degraded transcripts. Similarly, we calculated average coverage values in a window of 300nt around the coverage start and applied a similar strategy: 5' coverage values increased along the time course, confirming the accumulation of translationally suppressed mRNA species otherwise destined for degradation. Importantly, coverage values were normalized for each transcript, thus controlling for expression level changes (Methods). Also, we did not observe similar changes at the 3' end of transcripts (Appendix Fig. S13). Results were similar when using different cutoffs for the definition of coverage starting point (Appendix Fig. S14).

To test whether the suppression of 5'-3' decay of untranslated transcripts by DDX3X occurs in vivo, we re-analyzed recent RNA-seq/Ribo-seq dataset in a conditional *Ddx3x* (cKO) mouse model (Hoye et al, 2022) (Fig. 6), where cell cycle and neurogenesis defects are evident when *Ddx3x* is depleted in neuronal progenitors. After applying our analytical pipeline, we observed that the accumulation of untranslated transcripts is even more evident in this in vivo model, as is its relationship with *GCcds* values (Fig. 6A). Analogous to the strategy presented in Fig. 5, 5' coverage values, as well as coverage starting points (Appendix Fig. S15), differed significantly between wild type and *Ddx3x* cKO animals (Fig. 6B) in regulated transcripts, while no difference was detected at the 3'end (Appendix Fig. S16).

Leveraging again the power of hundreds of RNA-seq experiments, we examined 5' coverage profiles in the ENCODE dataset, partitioning experiments by their dependency on *GCcds* values. Differences between stabilized and control mRNAs are greater as the *GCcds* signature is more predominant (Fig. 6C). Aggregating different experiments according to their *GCcds* dependency for example transcripts (Fig. 6D) confirm this phenomenon, where both coverage starting position and coverage values changed across different datasets, indicative of mRNA decay regulation.

Taken together, we provide evidence for in vivo DDX3X-mediated stabilization of untranslated transcripts, its dependence on *GCcds* values, and, supporting the different analyses reported in this study (Fig. 7) a high-resolution RNA-seq coverage analysis strategy to investigate *GCcds*-related mRNA decay regulation, with support from hundreds of post-transcriptionally perturbed transcriptomes.

## Discussion

The multifaceted role of DDX3X, described as involved in different molecular processes, often hinders the ability to understand its functions, especially considering the interconnected nature of molecular processes in the cell. Multiple mRNA features might

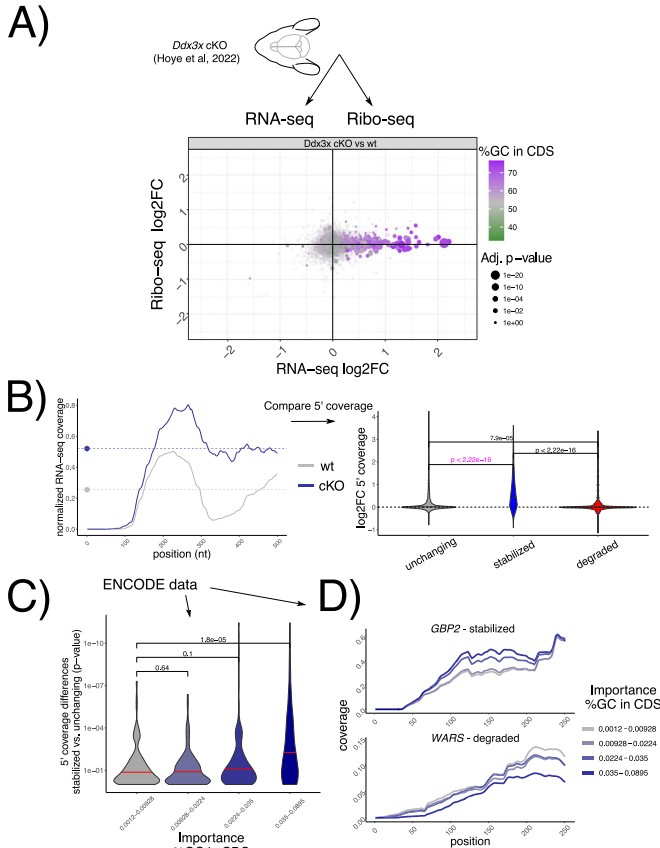

**Figure 6.   *GCcds* - mediated mRNA stabilization is detectable in vivo and across the ENCODE RBP database.**

(**A**) Changes in Ribo-seq and RNA-seq levels in a conditional Ddx3x mouse model, showing *GCcds* values. The size of the dots represents statistical significance from a differential translation efficiency test, as in Fig. 1A. (**B**) Strategy for coverage analysis in the mouse *Ddx3x* cKO experiment, shown for the *Ctxn1* gene. Differences in coverage values are extracted and compared across regulated mRNAs. In (**C**) the same strategy as in Fig. 5A applied to each differential analysis followed by RBP knockdown in the ENCODE dataset. Differences in coverage values between stabilized and unchanging mRNAs (shown by *p*-values, as calculated as in panel (**B**)), in pink color) are plotted against *GCcds* importance (*x* axis). Significance values from a one-sided Wilcoxon test. (**D**) Example mean coverage on 2 mRNAs (1 stabilized and 1 degraded), partitioning RBP knockdown datasets by their *GCcds* importance. An increase in coverage can be observed for the stabilized mRNA, while the opposite trend is visible for a degraded transcript.

underlie different modes of regulation, as we previously showed and experimentally validated 5'UTR dependencies underlying DDX3X translation regulation (Calviello et al, 2021). This outlines an unmet need for studies linking multiple aspects of the gene expression cascade.

In addition to profiling RNA levels and translation, we further dissected dynamics of cytoplasmic regulation by DDX3X, by employing a time course of efficient DDX3X depletion (Fig. 1A). Akin to previous studies observing translation suppression preceding mRNA changes during miRNA-mediated regulation (Bazzini et al, 2012), we observed an accumulation of translationally suppressed RNAs. This highlights the importance to profile not only mRNA abundance but also translation levels, which, in

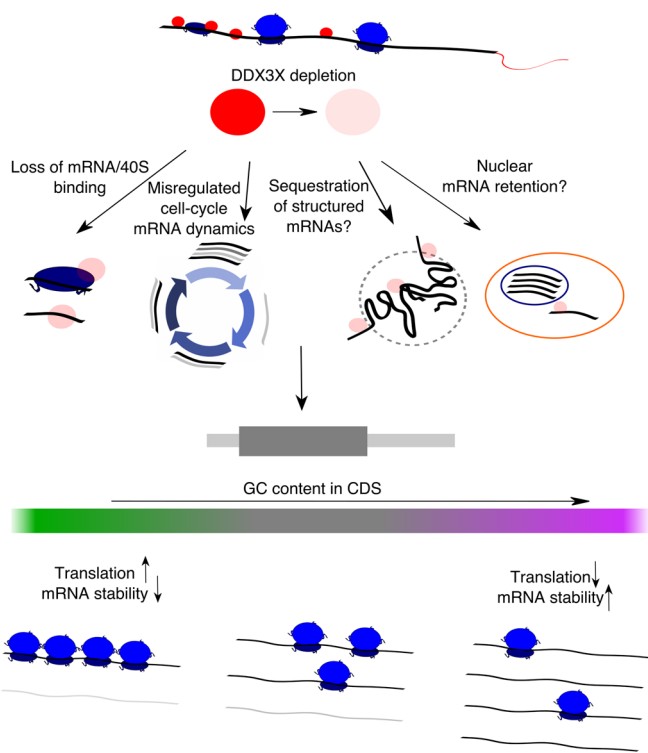

**Figure 7. A model for multimodal mRNA regulation by DDX3X.**

A schematic showing the effects of DDX3X depletion on GC-content related changes in translation and mRNA stability, highlighting potential molecular mechanisms underlying this phenomenon.

absence of quantitative estimates of regulated protein levels, can greatly help researchers understanding the functions of many cryptic regulators often involved in multiple processes, like DDX3X and other RBPs (Gerstberger et al, 2014). Despite relatively fast kinetics of DDX3X degradation from our degron system, more work needs to be performed to pinpoint exactly what changes occur right after DDX3X depletion, and to more precisely quantify the lag between translation suppression and mRNA stabilization.

By employing multiple techniques for feature selection, we identified a major feature underlying mRNA regulation by DDX3X, as well as by many other post-transcriptional regulators. An important area of investigation for the future is to employ more unbiased approaches, akin to recent Natural Language Processing-inspired methods in transcription regulation (Avsec et al, 2021), in mRNA biology to accurately estimate the relevant features directly from data rather than specified by potentially biased approaches. In our hands, the relevance of $GCcds$ is clearly picked up by both the Random Forest and the Lasso (Appendix Fig. S4). Importantly, we included similar features, such as overall GC content (Thomas et al, 2022), in UTRs, introns etc., alongside codon frequencies and previously estimated values of codon optimality (Hia et al, 2019).

Our study suggests that data-driven approaches to functional transcriptomics are highly needed, where data from multiple experiments are routinely re-analyzed to test hypotheses and provide new insights into the complex world of mRNA biology. However, while profiling translation allowed us to focus on specific mRNA classes and their features, no large-scale translation

profiling study exists yet, with few precious atlases recently appearing in the literature (Chothani et al, 2022). The current ENCODE RBP series is of great value to many mRNA biology researchers worldwide, and it has been an invaluable resource for many recent studies (Zhao et al, 2022; Van Nostrand et al, 2020b), yet an extension of these approaches which includes other aspects of post-transcriptional regulation, such as translation and stability, is in great need.

In the original ENCODE RBP study (Van Nostrand et al, 2020a), gene expression estimates were GC-corrected for each sample, as GC content has been often reported as a confounder, especially when comparing across sequencing technologies and labs. Given the presence of GC-related biases in sequencing-based assays, we think that great caution must be taken when observing expression changes driven by GC content features, especially when interpreted as direct effects from single molecular factors. Our degron time course analysis, despite containing dozens of features pertaining to GC content measures, detected GC content specifically in coding sequence as a feature underlying regulation, and this region-specific effect is not consistent with a general confounding role for $GCcds$. Moreover, our analysis focused on differences upon a perturbation under a single sequencing platform and laboratory settings, which are likely to have similar GC-related confounders, should there be any. Important confirmation of the relevance of $GCcds$ and its relationship to mRNA dynamics also came from: employing SLAM-seq to estimate differences in stability (Fig. 2), qPCR validations (Appendix Fig. S3), re-analysis of in vivo Ddx3x cKO RNA-seq/Ribo-seq (Fig. 6), re-analysis of hundreds of RBP perturbations in human cell lines (Fig. 4), and by analyzing kinetics extracted by transcriptome dynamics in cell-cycle specific states (Fig. 4).

Together with well-established differential analysis statistical methods, which allowed us to robustly identify different classes of regulated mRNAs, we exploited the high resolution offered by RNA-seq to analyze differences in 5'end coverage for thousands of individual transcripts (Fig. 5), as an additional metric reflecting active regulation of mRNA decay mechanisms. We posit that popular analysis strategies for -omics techniques, despite their popularity over more than a decade, often obscures information with regards to mRNA processing and other molecular mechanisms, which can be uncovered by dedicated computational methods. Importantly, such dynamics are invisible (or, worse, can significantly distort quantification estimates) when performing gene-level analyses.

The mechanism, or mechanisms, by which GC content in coding regions shapes mRNA dynamics is still to be determined. We speculate that complex RNA structures in the coding sequence can form in the absence of active translation elongation, and such structure may mediate degradation, helped by RNP complexes in the cytoplasm. However, recent literature focused on the role of different codons in mediating such effect (Medina-Muñoz et al, 2021). In our hands, codon-mediated effects seem to be negligible when considering the overall $GCcds$ values, but more work needs to be done to identify cases where one or the other, or a mix of the two, can mediate mRNA decay on different transcripts. The involvement of mRNA dynamics during the cell cycle (Fig. 4) suggests a model where, during cell cycle - dependent translation suppression, mRNAs are able to fold structures in the coding sequence promoting decay, and, when such processes are misregulated (e.g., by depleting multifunctional

RNA helicases such as DDX3X), this process is less efficient. The extent to which cell cycle changes might depend on direct DDX3X binding and regulation remains to be elucidated. Further work needs to be done to refine the exact function, together with the subcellular localization, of regulated mRNAs. For instance, mRNA retention in the nucleus might be an additional under-appreciated mode of gene expression control (Bahar Halpern et al, 2015), and is in line with our observation about the untranslated status of regulated transcripts. However, we identified GC content in the coding sequence as the hallmark feature for stabilized transcripts, a feature which is defined by translation in the cytoplasm.

While RBP binding data remains an important starting point from which we can build testable hypothesis, simple binding-to-function paradigms might also create bias when trying to explain complex phenotypes arising from RBP misfunction. Moreover, we observed how binding patterns might different between different regulated classes (Appendix Fig. S4). In our previous study we investigated the changes in translation and RNA abundance using a DDX3X helicase mutant; one of the observations we made pertained to the lack of RNA changes in our data, suggesting a potential function for the helicase activity in orchestrating such changes.

Previous work implicated DDX3X in mediating cell cycle dynamics by a variety of mechanisms (Kotov et al, 2016), including a direct regulation of cyclin E1 translation (Lai et al, 2010), which, however was not among the most regulated mRNAs in our dataset (Dataset EV2). More work needs to be done to accurately quantify mRNA dynamics and RBP functions in the cell cycle, where translation regulation mechanisms (Clemm von Hohenberg et al, 2022; Tanenbaum et al, 2015) ensure controlled rates protein synthesis. The connection between cell cycle, sequence content and mRNA regulation is reinforced by the in vivo data, adding to the importance of studying post-transcriptional regulation along the neurogenesis axis (Hoye and Silver, 2021; Harnett et al, 2022), where the equilibrium between proliferation, apoptosis and differentiation (Pilaz et al, 2016) shapes the complexity of the developing brain.

# Methods

## Reagents and tools

See Table 1.

**Table 1. Reagents and tools.**

| Reagent/Resource | Reference or Source | Identifier or Catalog Number |
|---|---|---|
| **Experimental Models** | | |
| HCT116 DDX3X-mAID | (Venkataramanan et al, 2021) | |
| **Antibodies** | | |
| anti-GAPDH | Rockland Immunochemicals | Cat# 600-401-A33S |
| Anti-DDX3 | (Calviello et al, 2021) | custom made by Genemed Synthesis using peptide ENALGLDQQFAGLDLNSSDNQS |
| **Oligonucleotides and other sequence-based reagents** | | |
| TaqMan probe RACK1 | Thermo Scientific | Chr.5: 181236928 – 181243906 - Hs00272002_m1 -VIC-MGB |
| TaqMan probe LGALS1 | Thermo Scientific | Chr.22: 37675606 – 37679802 - Hs00355202_m1 - VIC-MGB |
| TaqMan probe PFN1 | Thermo Scientific | Chr.17: 4945650 – 4949088 - Hs07291746_gH - VIC-MGB |
| TaqMan probe JUND | Thermo Scientific | Chr.19: 18279694 – 18281656 - Hs04187679_s1 - FAM-MGB |
| TaqMan probe EIF2A | Thermo Scientific | Chr.3: 150546678 – 150586016 - Hs00230684_m1 - FAM-MGB |
| **Chemicals, Enzymes and other reagents** | | |
| IAA (Indole-3-acetic acid) | Research Products International | Cat# I54000-5.0 |
| RNase I | Ambion | Cat# AM2294 |
| SUPERase | Ambion | Cat# AM2694 |
| T4 RNA Ligase 2 truncated KQ | NEB | Cat# M0373L |
| Superscript III | Invitrogen | |
| CircLigase II | Lucigen | Cat# CL4115K |
| **Software** | | |
| FACS DIVA | BD Life Sciences | |
| FlowJo V10 | BD Life Sciences | |
| bowtie2 | (Langmead et al, 2009) | |
| STAR | (Dobin et al, 2013) | |
| GenomicAlignments | (Huber et al, 2015) | |
| GenomicFiles | (Huber et al, 2015) | |

**Table 1.** (continued)

| Reagent/Resource | Reference or Source | Identifier or Catalog Number |
|---|---|---|
| INSPEcT | (De Pretis et al, 2015) | |
| RiboseQC | (Calviello et al, 2019) | |
| GenomicFeatures | (Lawrence et al, 2013) | |
| rtracklayer | (Lawrence et al, 2009) | |
| DESeq2 | (Love et al, 2014) | |
| randomForest | (Wiener, 2002) | |
| glmnet | (Friedman et al, 2010) | |
| **Other** | | |
| Illustra Microspin Columns S-400 HR | GE Healthcare | |
| Direct-zol kit | Zymo Research | |
| SLAMseq Kinetics Kit | Lexogen | |
| QuantSeq 3′ mRNA-Seq Library Prep Kit FWD for Illumina | Lexogen | |
| Click-iT™ Plus EdU Alexa Fluor™ 647 Flow Cytometry Assay Kit | Thermo Fisher | Cat# C10634 |
| FxCycle Violet DNA content stain | Thermo Fisher | Cat# F10347 |
| TruSeq Stranded Total RNA Human/Mouse/Rat kit | Illumina | |
| TaqMan® real-time PCR | Thermo Fisher | |

## Methods and protocols

### Ribo-seq and RNA seq experimental protocol

HCT116 cells with inducible degradation of DDX3X (as previously described (Venkataramanan et al, 2021)), were plated in 15 cm plates at 20% confluency (~$3.5 \times 10^6$ cells/plate). 48 hours post plating, when the cells were at ~70% confluency, the media was changed and fresh media with 500 µM IAA (Indole-3-acetic acid, the most common naturally occurring Auxin hormone) (Research Products International, cat: I54000-5.0) or DMSO was added to cells. Cells were harvested at 0, 4, 8, 16, 24, and 48 h post IAA addition. Cell number did not appreciably increase over the 48 hours of the experiment. To quantify DDX3X protein, we used an anti-DDX3X antibody described in previous work (Calviello et al, 2021) normalized to an anti-GAPDH antibody (Rockland Immunochemicals, cat: 600-401-A33S).

Cells were treated with 100 µg/ml cycloheximide (CHX), washed with PBS containing 100 µg/ml CHX, and immediately spun down and flash frozen. Once all time-points were collected, the cells were thawed and lysed in ice-cold lysis buffer (20 mM TRIS-HCl pH 7.4, 150 mM NaCl, 5 mM MgCl2, 1 mM DTT, 100 µg/ml CHX, 1% (v/v) Triton X-100, 25 U/ml TurboDNase (Ambion)). 240 µl lysate was treated with 6 µl RNase I (Ambion, 100 U/µl) for 45 min at RT with gentle agitation and further digestion halted by addition of SUPERase:In (Ambion). Illustra Microspin Columns S-400 HR (GE healthcare) were used to enrich for monosomes, and RNA was extracted from the flow-through using Direct-zol kit (Zymo Research). Gel slices of nucleic acids between 24 and 32 nts long were excised from a 15% urea-PAGE gel. Eluted RNA was treated with T4 PNK and preadenylated linker was ligated to the 3' end using T4 RNA Ligase 2 truncated KQ (NEB, M0373L).

Linker-ligated footprints were reverse transcribed using Superscript III (Invitrogen) and gel-purified RT products circularized using CircLigase II (Lucigen, CL4115K). rRNA depletion was performed using biotinylated oligos as described (Ingolia et al, 2012) and libraries constructed using a different reverse indexing primer for each sample.

For the RNA-seq, RNA was extracted from 25 µl intact lysate (non-digested) using the Direct-zol kit (Zymo Research) and stranded total RNA libraries were prepared using the TruSeq Stranded Total RNA Human/Mouse/Rat kit (Illumina), following manufacturer's instructions.

Libraries were quantified and checked for quality using a Qubit fluorimeter and Bioanalyzer (Agilent) and sequenced on a HiSeq 4000 sequencing system.

### SLAM-seq experimental protocol

SLAM-seq was performed at 60–70% confluency for DDX3X-mAID tagged HCT116. Media was changed and fresh media with 100 µM 4-thiouridine (4sU) was added to cells and changed every 3 h for 24 hours. 8 h prior to collection, growth medium was aspirated and replaced. Uridine chase was performed where cells were washed twice with 1× PBS and incubated with media containing 10 mM uridine and DMSO or 100 µM IAA for 0 or 8 h to induce degradation of DDX3X. At respective time points, cells were harvested followed by total RNA extraction using TRIzol (Ambion) following the manufacturer's instructions (SLAMseq Kinetics Kit – Catabolic Kinetics Module, Lexogen). Total RNA was alkylated by iodoacetamide for 15 min and RNA was purified by ethanol precipitation. 200 ng alkylated RNA were used as input for generating 3'-end mRNA sequencing libraries using a commercially available kit (QuantSeq 3′ mRNA-Seq Library Prep Kit FWD for Illumina, Lexogen).

### Ribo-seq data analysis

Reads were stripped of their adapter, collapsed, and UMI sequences were removed. Clean reads were then mapped to rRNA, tRNA, snoRNA and miRNA sequences using (Langmead et al, 2009) using sequences retrieved from UCSC browser and aligning reads were discarded. Remaining reads were mapped to the genome and transcriptome using STAR (Dobin et al, 2013) v2.7.9a supplied with the GENCODE v32 GTF file. STAR parameters were: *--outFilterMismatchNmax 3 --outFilterMultimapNmax 50 --chimScoreSeparation 10 --chimScoreMin 20 --chimSegmentMin 15 --outFilterIntronMotifs RemoveNoncanonicalUnannotated --alignSJoverhangMin 500 --outSAMmultNmax 1 --outMultimapperOrder Random.*

### SLAM-seq data analysis

Reads were mapped to the genome and transcriptome using same RNA-seq parameters, except for *--outFilterMismatchNmax 10*. Reads containing T > C mutations were extracted from the BAM file using *GenomicAlignments* and *GenomicFiles* Bioconductor (Huber et al, 2015) packages.

### RNA-seq data analysis

Reads were mapped to the genome and transcriptome using STAR with same Ribo-seq parameters. Synthesis, processing, and degradation rates were obtained using *INSPEcT* (De Pretis et al, 2015) v1.17, using default settings. Genes significantly changing in their dynamics at a *p*-value cutof of 0.05 were used for subsequent analysis.

### Differential analysis

Unique counts on different genomic regions were obtained using *RiboseQC* (Calviello et al, 2019). 5' end coverage values were inspected using Bioconductor packages such as *GenomicFeatures* (Lawrence et al, 2013) and *rtracklayer* (Lawrence et al, 2009). *DESeq2* (Love et al, 2014) was used to obtain RNA-seq, Ribo-seq, and TE regulation, as described previously (Calviello et al, 2021): changes in translation efficiency were calculated using *DESeq2* by using assay type (RNA-seq or Ribo-seq) as an additional covariate. Translationally regulated genes were defined using an FDR cutof of 0.05 from a likelihood ratio test, using a reduced model without the assay type covariate, e.g., assuming no difference between RNA-seq and Ribo-seq counts.

A similar strategy was used to define significant changes in DDX3X-mediated stability from SLAM-seq: count tables with T > C reads were built and analyzed using labeling (4sU/DMSO) and degron status (8 h. vs DMSO) as the two variables of interest; regulation in stability was defined using a reduced model without the degron type covariate, e.g. assuming no difference between DMSO and degron activation. Translationally regulated genes (as defined by Ribo-seq/RNA-seq) and stability regulated genes (as defined by SLAM-seq) were defined using a *p*-value cutof of 0.05.

For Figs. 1D and 3D, the coordinate system was divided into 70 bins on each axis. *GCcds* values (for Fig. 3D), or Ribo-seq and RNA-seq fold changes between each time point and the previous one (for Fig. 1D) were averaged across genes in the same bin. Only mRNAs with significant changes in translation efficiency at 48 h post degron induction were considered.

### Random Forest and Lasso regression

The Random Forest regression was run using the *randomForest* (Wiener, 2002) package with default parameters. Lasso regression was performed on scaled variables using the *glmnet* (Friedman et al, 2010) package. The entire input table is available in Dataset EV2, and a short description of the features follows:

TPM values using RNA-seq (in log scale). Baseline TE levels, defined as ratio of Ribo to RNA reads. Baseline RNA mature levels, defined as length-normalized ratio of RNA-seq reads in introns versus exons. GC content, length (in log scale) and Ribo-seq/RNA-seq density in: 5' UTRs, a window of 25nt around start and stop codons, CDS regions, non-coding internal exons, introns, and 3' UTRs. Codon frequencies. Measures of gene-specific codon optimality, previously calculated from a recent study (Medina-Muñoz et al, 2021). GC-content at first, second, or third codon position.

Feature importance (measured by mean decrease in accuracy for the random forest model and by the lasso coefficients) and correlation between predicted and measured test data were calculated on a 5-fold cross-validation scheme.

### Analysis of cell cycle-dependent mRNA dynamics

Estimated mRNA decay kinetics at cell cycle re-entry were deposited as supplementary files of the original study (Krenning et al, 2022). Genes were partitioned by dividing their *GCcds* values into three groups given the low number of quantified genes (total *n* = 220).

### Cell cycle staging

To measure DNA replication and cell cycle stage, EdU (5-ethynyl-2′-deoxyuridine) was added to cells at 10 nM for 1.5 h before harvesting. 1 confluent well of a 6-well plate of HCT116 cells were harvested and processed as per manufacturer's instructions for the Click-iT™ Plus EdU Alexa Fluor™ 647 Flow Cytometry Assay Kit (Thermo Fisher cat: C10634). Per manufacturer's instructions, FxCycle Violet DNA content stain (Thermo Fisher cat: F10347) was added after the Click-iT reaction at 1:1000 dilution before quantifying on a BD LSR Dual Fortessa flow cytometer. Alexa Fluor 647 was measured in the 670-30 Red C-A Channel and FxCycle Violet Stain was measured in the 450-50 Violet F-A Channel. Analysis was performed using FACS DIVA and FlowJo V10 (FlowJo, LLC) software.

### 5'end coverage analysis

Computation on single-nucleotide coverage values was performed using *rtracklayer* (Lawrence et al, 2009). For each differential analysis, we extracted the most 250 stabilized and the most 250 degraded genes ranking *p*-values from RNA-seq differential analysis. 1500 control RNAs were randomly sampled from non-regulated genes, using *p*-values >0.2 and TPM values > 3. Coverage values were 0–1 (min/max) normalized and the first position at value >0.15 was identified as coverage starting position. In addition, a general coverage starting point was selected by pooling all samples, and a window of 250nt around such position was used to calculate average coverage values around the coverage start. Log2 fold change with respect to the control condition were then calculated.

For degron data, starting position and log2FC coverage values were extracted and used as input for linear regression. For coverage values, intercept was omitted, as the first value was 0. Beta coefficients were then extracted and compared between stabilized, degraded, and control mRNAs.

For mouse *Ddx3x* cKO and ENCODE data, differences between starting position (knockdown vs wt) and log2FC (knockdown vs wt) in coverage values were used to compare stabilized, degraded and control mRNAs, bypassing the regression step (2 values were calculated, as only wt or knockdown conditions were present).

### TaqMan RT-PCR

DDX3X-mAID tagged HCT116 cells were plated in six-well plates at 30–40% confluency. 24 h post plating 500 μM IAA or DMSO was added to cells with or without 200 nM Actinomycin D (ActD). Total RNA was extracted from cells at 60–70% confluency using Direct-zol kit (Zymo Research) at 0 and 24 h post-ActD and IAA or DMSO treatment. TaqMan probes for JUND, EIF2A, RACK1, LGALS1, and PFN1 were pre-designed and purchased from ThermoFisher Scientific. Probes for degraded (EIF2A) or stabilized mRNAs (JUND) were conjugated with FAM dye while control mRNAs RACK1, LGALS1, and PFN1 were conjugated with VIC dye. For the TaqMan real-time quantitative PCR amplification reactions, we employed an Applied Biosystems QuantStudio 6 Real-Time PCR System instrument. Real-time PCR was conducted using TaqMan Fast Virus 1-Step Master Mix from Applied Biosystems in 384-well plates, following the manufacturer's protocol. Each well contained probes targeting mRNAs subject to degradation (EIF2A) or stabilization (JUND) along with controls (RACK1, LGALS1, or PFN1). All reactions were conducted in triplicate. Thermal cycling conditions adhered to the manufacturer's recommended standard protocol. The quantification of the target input amount was determined using the cycle threshold (CT) value, which corresponds to the point at which the PCR amplification plot crosses the threshold. Expression of degraded and stabilized mRNAs were normalized to each control.

## Datasets analyzed

ENCODE accession numbers can be found in Dataset EV3. *Ddx3x* knockout Ribo-seq and RNA-seq were analyzed from accession number GSE203078.

## Data availability

The datasets and computer code produced in this study are available in the following databases: Ribo-seq, RNA-seq and SLAM-seq data: Gene Expression Omnibus with accession GSE218433. Code to reproduce all figures and tables, together with processed data, is freely accessible on Github: https://github.com/calviellolab/DDX3X_GC_paper.

## Peer review information

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

## Acknowledgements

LC wants to thank Piero Angela (1928-2022) for inspiring him, and countless other kids in Italy, to pursue a scientific career. This work was supported by the National Institutes of Health DP2GM132932 and R01NS120667 (to SNF). SNF is a Pew Scholar in the Biomedical Sciences, supported by The Pew Charitable Trusts. ZJ was supported by the UCSF Moritz-Heyman Discovery Fellow Program. AX was supported by NIH F30 Ruth L Kirschstein National Research Service Award HD110250. ZJ and AX were supported by the UCSF Medical Scientist Training Program (T32GM007618). Flow cytometry data was generated at the UCSF Parnassus Flow CoLab (RRID:SCR_018206) and sequencing was performed at the UCSF CAT, supported by UCSF PBBR, RRP IMIA, and NIH 1S10OD028511-01 grants. Mouse image in Fig. 6A used with permission from https://doi.org/10.5281/zenodo.3925903. FD is a PhD student within the European School of Molecular Medicine (SEMM). This article was prepared while MLH was employed at Duke University. The opinions expressed in this article are the author's own and do not reflect the view of the National Institutes of Health, the Department of Health and Human Services, or the United States government.

## Author contributions

**Ziad Jowhar**: Validation; Investigation; Visualization; Methodology; Writing—review and editing. **Albert Xu**: Validation; Investigation; Visualization; Methodology; Writing—review and editing. **Srivats Venkataramanan**: Resources; Investigation; Visualization; Methodology; Writing—review and editing. **Francesco Dossena**: Data curation; Software; Validation; Visualization. **Mariah L Hoye**: Resources; Validation; Investigation; Methodology; Writing—review and editing. **Debra L Silver**: Supervision; Funding acquisition; Investigation; Writing—review and editing. **Stephen N Floor**: Conceptualization; Resources; Supervision; Funding acquisition; Investigation; Methodology; Writing—review and editing. **Lorenzo Calviello**: Conceptualization; Data curation; Software; Formal analysis; Supervision; Validation; Investigation; Visualization; Methodology; Writing—original draft; Writing—review and editing.

## Disclosure and competing interests statement

The authors declare no competing interests.

