## [Peer Review File · Molecular Systems Biology]

A ubiquitous GC content signature underlies multimodal mRNA regulation by DDX3X

Ziad Jowhar, Albert Xu, Srivats Venkataramanan, Francesco Dossena, Mariah Hoye, Debra Silver, Stephen Floor, and Lorenzo Calviello

DOI: 10.15252/msb.202312133

Corresponding author(s): Lorenzo Calviello (lorenzo.calviello@fht.org) , Stephen Floor (stephen.floor@ucsf.edu)

Review Timeline:

Submission Date:	27th Nov 23
Editorial Decision:	29th Nov 23
Revision Received:	10th Dec 23
Editorial Decision:	12th Dec 23
Revision Received:	21st Dec 23
Accepted:	3rd Jan 24

Editor: Maria Polychronidou

Transaction Report:

The reviewers' comments and authors' responses are not available with this article, as the initial review process took place with another journal.

29th Nov 2023

Manuscript Number: MSB-2023-12133

Title: A ubiquitous GC content signature underlies multimodal mRNA regulation by DDX3X

Dear Lorenzo,

Thank you again for submitting your work to Molecular Systems Biology. I have now read your manuscript, the referee reports from the other journal and your responses to these comments and discussed them with the team. Overall, we think that the review process has been constructive and therefore we can consider the study for publication using these reports, and without reviewing the study from scratch. We think that most technical and other core concerns seem to have been addressed in the previous round of revision at the other journal. Reviewers #1, #2 and #3 do not seem to have any remaining concerns that would preclude the publication of the study and I think that your responses to their concerns seem reasonable. Regarding the remaining concerns of reviewer #4 on clarifying the novelty and advance (especially over your previous work on DDX3X, Calviello et al, 2021) we think that the revised Discussion offers a balanced contextualization of the new findings over previous work. While, as mentioned by the reviewers and as discussed in your responses and in the manuscript itself, several mechanistic details regarding how DDX3X regulates RNA levels remain to be further explored, we feel that the presented findings are still relevant for the field. As such, we have decided to proceed with publishing the study in Molecular Systems Biology, pending some minor revisions, that would address the remaining minor concerns (e.g. those of reviewer #3 regarding figure presentation).

We would also ask you to fix some editorial issues listed below.

- Please provide 5 keywords.
- Please include a Disclosure and Competing Interests Statement in the main text.
- Please provide a .doc version of the manuscript text (including legends for main figures and tables) and individual files for the main figures. The figure legends should be included at the end of the manuscript text, after the References.
- The Reference list should be formatted according to the Molecular Systems Biology style i.e. ordered alphabetically and listing the first 10 authors followed by et al.
- We have replaced Supplementary Information by the Expanded View (EV format). In this case, all additional figures can be included in a PDF called Appendix. Appendix figures should be labeled and called out as: "Appendix Figure S1, Appendix Figure S2... Appendix Table S1..." etc. Each legend should be below the corresponding Figure/Table in the Appendix. Please include a Table of Contents in the beginning of the Appendix. For detailed instructions regarding expanded view please refer to our Author Guidelines: .
- Supplementary Tables (if they are shorter than one page) should be included in the Appendix and renamed to Appendix Tables. More complex tables should be provided as EV datasets (either as .xls files or .zip folders). Please provide one file per EV Dataset. Please include the description of each EV Dataset in the dataset file itself, ie. in a separate tab for .xls files or as a README.txt file in .zip folders.
- Please provide a "standfirst text" summarizing the study in one or two sentences (approximately 250 characters), three to four "bullet points" highlighting the main findings and a "synopsis image" (550px width and max 400px height, jpeg format) to highlight the paper on our homepage.
- All Materials and Methods need to be described in the main text. We would encourage you to use 'Structured Methods', our new Materials and Methods format. According to this format, the Material and Methods section should include a Reagents and Tools Table (listing key reagents, experimental models, software and relevant equipment and including their sources and relevant identifiers) followed by a Methods and Protocols section in which we encourage the authors to describe their methods using a step-by-step protocol format with bullet points, to facilitate the adoption of the methodologies across labs. More information on how to adhere to this format as well as downloadable templates (.doc or .xls) for the Reagents and Tools Table can be found in our author guidelines: . An example of a Method paper with Structured Methods can be found here:
- Please include a Data availability section describing how the data, code etc. generated in this study have been made available. This section needs to be formatted according to the example below:
The datasets and computer code produced in this study are available in the following databases:
 - Chip-Seq data: Gene Expression Omnibus GSE46748 (<https://www.ncbi.nlm.nih.gov/geo/query/acc.cgi?acc=GSE46748>)
 - [data type]: [full name of the resource] [accession number/identifier] ([doi or URL or identifiers.org/DATABASE:ACCESSION])

- The Data Availability section should be reserved only for data generated in the study. Data retrieved from other sources (e.g. previously published data) should be referenced/described in a separate section titled "Datasets analysed".

- Molecular Systems Biology supports formal data citations in the Reference list, to cite previously published datasets. In addition to citing the original papers that reported the data, we encourage you to also cite the relevant datasets directly in the Reference list. In the text, references to datasets are included as "Data ref: Smith et al, 2001" or "Data ref: NCBI Sequence Read Archive PRJNA342805, 2017". In the Reference list, data citations are very similar to normal literature references but must be labeled with "[DATASET]" at the end of the reference. For detailed instructions please refer to our Author Guidelines .

- When you resubmit your manuscript, please download our CHECKLIST (<https://bit.ly/EMBOPressAuthorChecklist>) and include the completed form in your submission. *Please note* that the Author Checklist will be published alongside the paper as part of the transparent process (<https://www.embopress.org/page/journal/17444292/authorguide#transparentprocess>)

Please resubmit your revised manuscript online, with a covering letter listing amendments and responses to each point raised by the referees. Please resubmit the paper ****within one month**** and ideally as soon as possible. If we do not receive the revised manuscript within this time period, the file might be closed and any subsequent resubmission would be treated as a new manuscript. Please use the Manuscript Number (above) in all correspondence.

Click on the link below to submit your revised paper.

Kind regards,

Maria

Maria Polychronidou, PhD
Senior Editor
Molecular Systems Biology

If you do choose to resubmit, please click on the link below to submit the revision online before 29th Dec 2023.

IMPORTANT: When you send your revision, we will require the following items:

1. the manuscript text in LaTeX, RTF or MS Word format
2. a letter with a detailed description of the changes made in response to the referees. Please specify clearly the exact places in the text (pages and paragraphs) where each change has been made in response to each specific comment given
3. three to four 'bullet points' highlighting the main findings of your study
4. a short 'blurb' text summarizing in two sentences the study (max. 250 characters)
5. a 'thumbnail image' (550px width and max 400px height, Illustrator, PowerPoint or jpeg format), which can be used as 'visual title' for the synopsis section of your paper.
6. Please include an author contributions statement after the Acknowledgements section (see <https://www.embopress.org/page/journal/17444292/authorguide#manuscriptpreparation>)
7. Please complete the CHECKLIST available at (<https://bit.ly/EMBOPressAuthorChecklist>). Please note that the Author Checklist will be published alongside the paper as part of the transparent process (<https://www.embopress.org/page/journal/17444292/authorguide#transparentprocess>).
8. When assembling figures, please refer to our figure preparation guideline in order to ensure proper formatting and readability in print as well as on screen:

See also figure legend guidelines: <https://www.embopress.org/page/journal/17444292/authorguide#figureformat>

9. Please note that corresponding authors are required to supply an ORCID ID for their name upon submission of a revised manuscript (EMBO Press signed a joint statement to encourage ORCID adoption).

(<https://www.embopress.org/page/journal/17444292/authorguide#editorialprocess>)

Currently, our records indicate that the ORCID for your account is 0000-0002-5600-0988.

Link Not Available

*** PLEASE NOTE *** As part of the EMBO Press transparent editorial process initiative (see our Editorial at <https://dx.doi.org/10.1038/msb.2010.72> , Molecular Systems Biology will publish online a Review Process File to accompany accepted manuscripts. When preparing your letter of response, please be aware that in the event of acceptance, your cover letter/point-by-point document will be included as part of this File, which will be available to the scientific community. More information about this initiative is available in our Instructions to Authors. If you have any questions about this initiative, please contact the editorial office (msb@embo.org).

12th Dec 2023

Manuscript Number: MSB-2023-12133R

Title: A ubiquitous GC content signature underlies multimodal mRNA regulation by DDX3X

Dear Lorenzo,

Thank you for sending us your revised manuscript. We think that all remaining issues of the reviewers have now been addressed and we can accept the manuscript for publication, pending some final editorial requests listed below.

- Our data editors have noted that the following needs to be edited in the figure legends:

-- Please define the annotated p values ****/** in the legend of figures 2b; 3f; as appropriate.

-- Please indicate the statistical test used for data analysis in the legends of figures 1a; 3a; 4d; 6a-c.

-- Please note that the box plot needs to be defined in terms of minima, maxima, center, bounds of box and whiskers, and percentile in the legend of figure 4e.

-- Please note that information related to n is missing in the legends of figures 2a; 3b-c, e; 5; 6b-c.

-- Although 'n' is provided, please describe the nature of entity for 'n' in the legends of figures 2b; 3f; 4e-f.

-- Please define the error bars in the legends of figures 2a-b; 3b-c, e-f; 4f.

- The funding information provided in the manuscript text need to match the information entered in the online submission system. Currently the following is missing from the submission system: The Pew Charitable Trusts; the UCSF Moritz-Heyman Discovery Fellow Program; NIH F30 Ruth L Kirschstein National Research Service Award HD110250; the UCSF Medical Scientist Training Program (T32GM007618); UCSF PBBR, RRP IMIA, and NIH 1S10OD028511-01

- The References should be formatted according to the Molecular Systems Biology reference style (i.e., ordered alphabetically and listing the first 10 authors followed by et al).

- Please remove the 'Authors Contributions' from the manuscript. The 'Author Contributions' section is replaced by the CRediT contributor roles taxonomy to specify the contributions of each author in the journal submission system. Please use the free text box in the 'author information' section of the online submission system to provide more detailed descriptions if needed (e.g., 'X provided intracellular Ca⁺⁺ measurements in fig Y').

Please resubmit your revised manuscript online ****within two weeks**** and ideally as soon as possible. If we do not receive the revised manuscript within this time period, the file might be closed and any subsequent resubmission would be treated as a new manuscript. Please use the Manuscript Number (above) in all correspondence.

Click on the link below to submit your revised paper.

Kind regards,

Maria

Maria Polychronidou, PhD
Senior Editor
Molecular Systems Biology

If you do choose to resubmit, please click on the link below to submit the revision online before 11th Jan 2024.

IMPORTANT:

Please note that corresponding authors are required to supply an ORCID ID for their name upon submission of a revised

manuscript (EMBO Press signed a joint statement to encourage ORCID adoption).
(<https://www.embopress.org/page/journal/17444292/authorguide#editorialprocess>)
Currently, our records indicate that the ORCID for your account is 0000-0002-5600-0988.

Please click the link below to modify this ORCID:
Link Not Available

*** PLEASE NOTE *** As part of the EMBO Press transparent editorial process initiative (see our Editorial at <https://dx.doi.org/10.1038/msb.2010.72> , Molecular Systems Biology will publish online a Review Process File to accompany accepted manuscripts. When preparing your letter of response, please be aware that in the event of acceptance, your cover letter/point-by-point document will be included as part of this File, which will be available to the scientific community. More information about this initiative is available in our Instructions to Authors. If you have any questions about this initiative, please contact the editorial office (msb@embo.org).

All editorial and formatting issues were resolved by the authors.

3rd Jan 2024

Manuscript number: MSB-2023-12133RR

Title: A ubiquitous GC content signature underlies multimodal mRNA regulation by DDX3X

Dear Lorenzo,

Thank you again for sending us your revised manuscript. We are now satisfied with the modifications made and I am pleased to inform you that your paper has been accepted for publication.

Best wishes and Happy New year!

Maria

Maria Polychronidou, PhD
Senior Editor
Molecular Systems Biology
